# Evaluating the Protective Effect of Intravesical Bacillus Calmette-Guerin against SARS-CoV-2 in Non-Muscle Invasive Bladder Cancer Patients: A Multicenter Observational Trial

**DOI:** 10.3390/cancers15051618

**Published:** 2023-03-06

**Authors:** Rodolfo Hurle, Francesco Soria, Roberto Contieri, Pier Paolo Avolio, Stefano Mancon, Massimo Lazzeri, Valentina Bernasconi, Simone Mazzoli, Giuseppe Pizzuto, Matteo De Bellis, Matteo Rosazza, Simone Livoti, Tommaso Lupia, Silvia Corcione, Beatrice Lillaz, Francesco Giuseppe De Rosa, Nicolò Maria Buffi, Ashish M. Kamat, Paolo Gontero, Paolo Casale

**Affiliations:** 1Department of Urology, IRCCS Humanitas Research Hospital, Via Manzoni 56, 20089 Rozzano, Italy; 2Division of Urology, Department of Surgical Sciences, San Giovanni Battista Hospital, Torino School of Medicine, 10126 Turin, Italy; 3Department of Biomedical Sciences, Humanitas University, Via Rita Levi Montalcini 4, 20090 Pieve Emanuele, Italy; 4Department of Medical Sciences, Infectious Diseases, University of Turin, A.O.U. Città della Salute e della Scienza di Torino, 10126 Turin, Italy; 5MD Anderson Cancer Center, University of Texas, Houston, TX 78712, USA

**Keywords:** non-muscle invasive bladder cancer, Bacillus Calmette-Guerin, SARS-CoV-2 infection

## Abstract

**Simple Summary:**

The transurethral resection of bladder tumors followed by intravesical Bacillus Calmette-Guerin (BCG) instillations represents the standard treatment for high-risk and selected intermediate-risk non-muscle invasive bladder cancer (NMIBC) patients. We hypothesized that intravesical BCG might be protective against symptomatic SARS-CoV-2 infection, especially in those patients who experienced systemic adverse events during BCG treatment. We tested our hypothesis in a large multicenter cohort of NMIBC patients treated with adjuvant intravesical BCG in the year preceding the first and second waves of the SARS-CoV-2 pandemic at two tertiary urological centers in Northern Italy.

**Abstract:**

We aim to evaluate the potential protective role of intravesical Bacillus Calmette-Guerin (BCG) against SARS-CoV-2 in patients with non-muscle invasive bladder cancer (NMIBC). Patients treated with intravesical adjuvant therapy for NMIBC between January 2018 and December 2019 at two Italian referral centers were divided into two groups based on the received intravesical treatment regimen (BCG vs. chemotherapy). The study’s primary endpoint was evaluating SARS-CoV-2 disease incidence and severity among patients treated with intravesical BCG compared to the control group. The study’s secondary endpoint was the evaluation of SARS-CoV-2 infection (estimated with serology testing) in the study groups. Overall, 340 patients treated with BCG and 166 treated with intravesical chemotherapy were included in the study. Among patients treated with BCG, 165 (49%) experienced BCG-related adverse events, and serious adverse events occurred in 33 (10%) patients. Receiving BCG or experiencing systemic BCG-related adverse events were not associated with symptomatic proven SARS-CoV-2 infection (*p* = 0.9) nor with a positive serology test (*p* = 0.5). The main limitations are related to the retrospective nature of the study. In this multicenter observational trial, a protective role of intravesical BCG against SARS-CoV-2 could not be demonstrated. These results may be used for decision-making regarding ongoing and future trials.

## 1. Introduction

The emergence of a novel coronavirus in late 2019, the severe acute respiratory syndrome coronavirus 2 (SARS-CoV-2), rapidly turned into a dramatic global pandemic.

However, the recent advent of effective vaccines against SARS-CoV-2 seems to have mitigated the effects of the pandemic.

Bacillus Calmette-Guerin (BCG) is a vaccine that was developed in 1921 to provide immunity against tuberculosis, a bacterial infection that primarily affects the lungs. The vaccine is produced by attenuating a strain of Mycobacterium bovis, which is a bacterium closely related to the one that causes tuberculosis. The attenuated bacteria used in BCG are weakened to the point that they cannot cause disease, but are still able to stimulate the immune system to produce a response against tuberculosis [1].

Since its introduction, BCG has become one of the most widely used vaccines worldwide, especially in countries with high tuberculosis rates. One of the remarkable benefits of the BCG vaccine is that it not only protects against tuberculosis, but also against other infectious diseases. This has been observed since the beginning of the last century, when studies showed that BCG vaccination was associated with a significant reduction in infant mortality rates [2].

The reason for this more general protection has yet to be fully understood, but it is believed that the immune response triggered by the BCG vaccine provides a level of non-specific protection against other infectious agents. This is thought to occur because the BCG vaccine induces a robust activation of the immune system, leading to the production of cytokines and other immune mediators that enhance the ability of the immune system to combat a wide range of pathogens [3].

In summary, the BCG vaccine has been shown to be highly effective in preventing tuberculosis, and also provides a level of non-specific protection against other infectious diseases. This has made the BCG vaccine a valuable tool in public health, particularly in regions with highly prevalent infectious diseases [3].

Indeed, evidence from the beginning of the last century demonstrates that BCG vaccination could reduce infant mortality by up to 50%, not only as a direct consequence of the induced immune response against Mycobacterium tuberculosis, but also due to more general protection against unrelated infectious agents [3].

Since then, the cross-reactivity of BCG has been further investigated.

BCG has been shown to reduce the level of yellow fever vaccine viremia after vaccination through the induction of cytokine responses, with a crucial role for IL-1B [4]. Moreover, BCG vaccination before influenza vaccination resulted in a more pronounced increase and accelerated induction of immune response against the H1N1 vaccine strain [5]. The phase III ACTIVATE trial aimed to assess the efficacy of the BCG vaccine in diminishing the incidence of new infections in the elderly [6]; the interim analysis revealed a 53% decrease in new infections and an 80% decrease in respiratory tract infections in the BCG group.

Based on these considerations and on the observation that an inverse correlation between BCG vaccination coverage and SARS-CoV-2-associated morbidity and mortality has been reported in countries such as Japan [7], several trials assessing the potential protective role of BCG vaccination against SARS-CoV-2 have been initiated.

Intravesical BCG represents the standard treatment for high-risk and selected intermediate-risk non-muscle invasive bladder cancer (NMIBC) patients [8]. Unfortunately, side effects during BCG treatment are not unusual. While most patients experience none or mild events such as symptoms of cystitis, hematuria and general malaise with transient fever, a small percentage of patients develop severe adverse events (mainly persistent high-grade fever, arthralgia and arthritis) as a consequence of BCG hematogenous dissemination.

Therefore, we hypothesized that intravesical BCG might be protective against symptomatic SARS-CoV-2 infection, especially in those patients who experienced systemic adverse events during BCG treatment. We tested our hypothesis in a large multicenter cohort of NMIBC patients treated with adjuvant intravesical BCG in the year preceding the first and second waves of the SARS-CoV-2 pandemic at two tertiary urological centers in Northern Italy.

## 2. Materials and Methods

### 2.1. Study Population and Study Design

We report results from a multicenter observational review board-approved study (00174/2020). Consecutive patients treated with intravesical adjuvant therapy for NMIBC between January 2018 and December 2019 at two Italian referral centers were included in the study. Enrolled patients were divided into two groups based on the received intravesical treatment regimen: those treated with intravesical BCG (BCG seed RIVM (Medac^®^, D-20354 Hamburg, Germany; 2 × 108–3 × 109 CFU) (study group) and those treated with intravesical chemotherapy, this latter group serving as the control. The choice to select the control group among the bladder cancer population was made to minimize the risk of selection bias and retrieve as homogeneous a study population as possible (concerning median age, gender and lifestyle). According to international guidelines and recommendations, intravesical BCG was administered to high-risk and some intermediate-risk patients. In addition, intravesical adjuvant chemotherapy was administered in case of intermediate-risk disease, while single postoperative instillation of chemotherapy was given to low-risk patients.

Variables collected included baseline demographic characteristics and those inherent to BCG treatments (number of instillations, maintenance scheme and tolerability profile).

The study was conducted in three different steps:A phone interview was conducted among the study population between May 2020 and September 2020 (after the so-called “first SARS-CoV-2 wave”). Patients were asked to answer an ad hoc survey regarding SARS-CoV-2 infection previously built by the Infectious Disease Team (TL, SC and FDR).A serology test to highlight the presence of direct antibodies against COVID-19 (meaning, therefore, previous exposure to the virus) was offered to all patients who tested negative (molecular test) or had never tested for SARS-CoV-2 infection.Due to the occurrence of a second wave (greater than the first) of disease spread in Italy during autumn and winter 2020, all patients were again reached by phone and their profile of exposure to SARS-CoV-2 was updated.

Patients who did not answer the survey were excluded from the study.

### 2.2. Endpoints

The study’s primary endpoint was the evaluation of SARS-CoV-2 disease incidence and severity among patients treated with intravesical BCG compared to the control group. The study’s secondary endpoint was the evaluation of SARS-CoV-2 infection (estimated with serology testing) among the study groups.

### 2.3. Statistical Analysis

Absolute numbers and proportions were used to describe categorical variables, while median and interquartile ranges (IQR) were used for continuous variables. Chi-square, Fisher exact and Kruskal–Wallis tests were used when appropriate to compare the populations. Logistic regression models were built to evaluate the predictive role of intravesical BCG in preventing SARS-CoV-2 disease and infection. Statistical analyses were performed using STATA 16 (Stata Corp., College Station, TX, USA). All tests were two-sided, and *p* < 0.05 was considered statistically significant.

## 3. Results

Patients’ baseline characteristics are listed in Table 1.

Overall, 506 patients with NMIBC treated with adjuvant intravesical therapy were included in the study. Of these, 340 (67%) received intravesical BCG while 166 (33%) were treated with intravesical chemotherapy. Among the 340 patients treated with BCG, 165 (49%) experienced BCG-related adverse events (Table 2).

The most-frequently reported adverse events were symptoms of cystitis, fever/general malaise and hematuria. Serious adverse events, possible expression of BCG dissemination, such as arthritis and high-grade persistent fever occurred in 19 (6%) and 14 (4%) patients, respectively.

Among the patients treated with BCG, 185 (54%) reported symptoms consistent with possible SARS-CoV-2 infection (mainly flu-like symptoms and fever); however, this was confirmed with a positive molecular test in only 8 patients (2%). Similarly, 73 patients (44%) treated with intravesical chemotherapy experienced SARS-CoV-2-like symptoms and a SARS-CoV-2 infection was confirmed in 4 (2%) of them (Table 3).

Overall, 320 patients (67%) underwent a serology test to highlight the presence of direct antibodies against SARS-CoV-2. Of these, 214 were treated with BCG and 104 with intravesical chemotherapy. A positive serology test was found in 15 patients (7%) of the BCG group and in 9 patients (9%) of the chemotherapy group (*p* = 0.6).

Receiving BCG or experiencing systemic BCG-related adverse events were not associated with symptomatic proven SARS-CoV-2 infection (OR 0.98, 95% CI 0.29–3.29, *p* = 0.9) nor with a positive serology test (OR 0.77, 95% CI 0.37–1.61, *p* = 0.5).

## 4. Discussion

The outbreak of the COVID-19 epidemic had a tremendous influence on the management of cancer [9]. The BCG is a live attenuated vaccine that represents the most widely used vaccine in the world, assuring over 50% protection against lung respiratory diseases and over 80% protection against tuberculosis [10]. Numerous studies have documented the BCG vaccine’s cross-protective benefits against diseases unrelated to tuberculosis [11]. The hypothesis of a protective role of BCG towards SARS-CoV-2 comes from multiple sources of evidence. First, the BCG vaccine has been shown to induce non-specific effects on the immune system, thus protecting against a wide range of infections other than tuberculosis. In three randomized controlled trials in Guinea-Bissau, the BCG vaccine was administered to low-weight neonates to reduce infant mortality rates, with an observed beneficial effect in the neonatal period [12,13,14]. A meta-analysis of the three trials showed that BCG reduced mortality by 38% within the neonatal period and 16% by the age of 12 months, mainly due to reduced infectious disease mortality [14]. Second, the BCG vaccine reduced yellow fever vaccine viremia (a single-stranded positive-sense RNA virus such as SARS-CoV-2) by 71% in humans and reduced the severity of mengovirus infection in mice [15]. The ability of BCG to enhance the protection against unrelated infectious agents calls into question multiple mechanisms, such as the molecular similarity between BCG antigens and some viral antigens, the so-called heterologous immunity leading to the activation of bystander B and T cells, and the trained immunity resulting in an increasingly active immune response [16,17].

Early evidence from the current SARS-CoV-2 pandemic highlighted different incidences and severities of disease across different countries, probably due to differences in genetic susceptibility, cultural behaviors, mitigation norms and healthcare systems. However, it has been proposed that a partial explanation of these differences may rely on different national policies regarding BCG vaccination [18]. According to several epidemiological studies, the incidence of SARS-CoV-2 is four times higher in countries without universal BCG vaccination than those with this policy [19,20,21]. However, as correctly highlighted by Desouky [18], observation/correlation does not mean causation. To fill this gap, several studies aiming to test the efficacy of BCG vaccination against SARS-CoV-2 in different populations such as healthcare workers or the elderly population have been recently published.

In a retrospective study, BCG revaccination was shown to be protective against COVID-19 infections in high-risk healthcare workers. Specifically, none of the patients who received the BCG booster vaccination developed COVID-19 infection, compared to 8.6% in the unvaccinated group [22].

In contrast, the results from a large cohort of Israeli adults who chose or chose not to receive the BCG vaccination in childhood did not show differences in the incidence of COVID-19 [23],

Several randomized trials investigating the protective effect of BCG vaccination against SARS-CoV-2 were registered [24]. The BADAS trial (NCT04348370), initiated in April 2020, aimed to randomize 1800 healthcare workers to receive BCG vaccination or placebo. The primary endpoint of the study was the incidence of SARS-CoV-2 infection and disease severity.

The ACTIVATE-2 study was a multicenter, double-blind trial that randomized 301 volunteers aged >50 to receive vaccination with BCG versus placebo. The primary end points were the incidence of COVID-19 and the presence of anti–SARS-CoV-2 antibodies. At 6 months, individuals vaccinated with BCG showed a lower incidence of COVID-19 (OR 0.32 95% CI 0.13–0.79, *p* = 0.014) [25].

In contrast, in a unicentric randomized phase II clinical trial, Dos Anjos et al. did not find a statistically significant lower rate of incidence of COVID-19 positivity in healthcare workers revaccinated with M. bovis BCG Moscow [26].

Similarly, a Dutch multicentric randomized trial compared the number of days of unplanned absenteeism for any reason during the COVID-19 pandemic in healthcare workers randomized to receive BCG vaccination or placebo. Again, no protective role of BCG vaccination emerged [27].

A third randomized double-blind placebo-controlled trial enrolling healthcare workers found that BCG vaccination did not have a protective role against COVID19 infection or symptoms [28].

Lastly, the protective role of the genetically modified BCG vaccine VPM1002 was evaluated in a phase III randomized double-blind placebo-controlled multicenter clinical trial. VPM1002 is a modified BCG vaccine with improved immunogenicity and safety profile. However, although the authors reported a lower number of days with severe RTI in the elderly vaccinated with VPM1002, they did not find a statistically significant difference between groups [29].

It should be considered the cited trials only focused on the role of BCG vaccination. The urological community has been using intravesical BCG as standard adjuvant treatment for patients with high-risk NMIBC since 1970 [30]. Despite the intravesical administration of BCG, some is absorbed and is able to induce systemic effects. Within 2–8 h of intravesical BCG instillation, a peak of cytokines and chemokines leading to the recruitment of immune cells to the bladder can be observed.

Moreover, intravesical BCG stimulates the humoral immune response by increasing IgG levels against tuberculin and mycobacterial heat shock proteins [18,31]. Finally, more than 40% of patients receiving intravesical BCG experience conversion of a previously negative tuberculin skin test [32]. Therefore, there is evidence to support the systemic immunological impact of intravesical BCG.

Patients treated with intravesical BCG showed a lower fatality rate (death/cases with respect to overall population) in a small retrospective Chilean study. However, the results are limited by the weak study design [33].

Contrastingly, Karabay et al. compared the incidence of SARS-CoV-2 infection in bladder cancer patients treated with or without intravesical BCG; in this study, no differences emerged between groups [34].

Finally, no evidence of BCG’s protective effect was shown in a recent retrospective study that included 2803 patients treated with intravesical BCG in an Italian region [35].

In recent years, several vaccines targeting the SARS-CoV-2 virus have been developed [36]. Although these vaccines have demonstrated high efficacy and have had a substantial positive impact on mitigating the pandemic, the durability of their protective effect over the long term remains uncertain. Investigating the potential adjunctive role of BCG enhancing the immune response to COVID-19 vaccination may be an area of interest for future research.

In this multicenter observational trial, a protective role of intravesical BCG against SARS-CoV-2 infection and disease could not be demonstrated. However, we found that patients treated with intravesical BCG for NMIBC harbor the same risk of contracting the SARS-CoV-2 infection and developing symptomatic disease as patients who did not receive intravesical immunotherapy. These findings were also confirmed in the subgroup of patients who experienced severe BCG-related adverse events due to a hematogenous BCG dissemination. These findings may be used to guide decision-making regarding ongoing and future trials aiming to explore the role of BCG in the prevention of SARS-CoV-2 infection.

Despite the novelty and significance of our study, it is important to note that it is not without limitations. Perhaps the most significant of these limitations is the inherent observational nature of the study design, which prevents us from drawing causal conclusions about the relationships we observe between intravesical BCG and SARS-CoV-2.

Another limitation of our study is related to the testing for SARS-CoV-2. First, we were not able to test all included patients for SARS-CoV-2, as the choice to undergo a serology test was left to the discretion of the patients. This introduces a potential selection bias. Furthermore, it is important to note that the serology tests were performed after the so-called “first wave” of the pandemic. As a result, it is possible that a higher number of patients may have contracted SARS-CoV-2 in an asymptomatic form during the second and third waves of the pandemic, with a possible impact on the results of the study. This temporal limitation may have led to an underestimation of the true prevalence of SARS-CoV-2 infection in the studied population, which could affect the accuracy of our findings.

Despite these limitations, our study provides valuable insights into the potential role of intravesical BCG and the risk of SARS-CoV-2 infection. However, it is important to acknowledge these limitations and the need for additional studies to further elucidate the relationship between these variables and SARS-CoV-2 infection.

## 5. Conclusions

In this multicenter observational trial, which aimed to investigate the potential protective effect of intravesical BCG against SARS-CoV-2, the data did not demonstrate a significant protective effect. While the results are not definitive, they suggest that intravesical BCG is unlikely to be a reliable strategy for preventing SARS-CoV-2 infection.

These results may have implications for decision-making regarding ongoing trials and future studies that might explore the adjunctive role of BCG in enhancing the immune response to COVID-19 vaccination. However, it is important to note that additional studies are needed to fully elucidate the potential of BCG and other agents in this regard.

## Figures and Tables

**Table 1 cancers-15-01618-t001:** Descriptive characteristics for the cohort of 506 patients with non-muscle invasive bladder cancer treated with adjuvant intravesical therapy between January 2018 and December 2019.

Variables	Total	Type of Intravesical Treatment		*p*-Value
		BCG	Chemotherapy	
Number of patients, n (%)	506	340 (67)	166 (33)	
Median age (IQR), years	73 (67–79)	74 (68–80)	72 (64–78)	0.05
Gender, n (%) Female Male	87 (17) 419 (83)	51 (15) 289 (85)	36 (22) 130 (78)	0.06
Median number of BCG induction instillations (IQR)	-	6 (6-6)	-	-
Type of adjuvant chemotherapy treatment, n (%)				-
Induction cycle	-	-	138 (83)	
Single postoperative instillation	-	-	28 (17)	

**Table 2 cancers-15-01618-t002:** Adverse events reported during intravesical treatment with Bacillus Calmette-Guerin among the study group (n = 340).

Symptoms	Frequency, n (%)
None	175 (51)
Cystitis	112 (33)
Hematuria	63 (19)
Epididymitis	9 (3)
Fever/general malaise	81 (24)
Arthralgia/arthrititis	19 (6)
High-grade persistent fever	14 (4)
Symptoms requiring treatment	88 (26)

**Table 3 cancers-15-01618-t003:** SARS-CoV-2-like symptoms and SARS-CoV-2 disease characteristics among the study population. Symptoms are not exclusive; a patient may have developed more than one symptom.

SARS-CoV-2-like Symptoms, n (%)	BCG (n = 340)	Chemotherapy (n = 166)	*p* Value
Flu-like symptoms in the last 90 days	53 (16)	24 (15)	0.8
Fever	30 (9)	8 (5)	0.1
Cough	26 (8)	10 (6)	0.5
Dry cough	9 (3)	8 (5)	0.2
Shortness of breath	7 (2)	2 (1)	0.5
Asthenia	17 (5)	6 (4)	0.4
Myalgia/arthralgia	8 (2)	5 (3)	0.7
Headache	4 (1)	3 (2)	0.6
Diarrhoea	7 (2)	1 (1)	0.2
Nausea/vomiting	7 (2)	3 (2)	0.6
Symptoms requiring hospitalization	4 (1)	1 (1)	0.6
Symptoms requiring medical examination	13 (4)	2 (1)	0.1
**SARS-CoV-2 Disease, n (%)**	**BCG (n = 340)**	**Chemotherapy (n = 166)**	***p* value**
Contact with SARS-CoV-2-positive patients Molecular test for suspected SARS-CoV-2 infection Positive molecular test SARS-CoV-2 disease requiring hospitalization Length of stay, days SARS-CoV-2 pneumonia SARS-CoV-2 requiring intensive care unit	16 (5) 23 (7) 8 (2) 1 (0) 10 1 (0) 0	12 (7) 14 (8) 4 (2) 1 (1) 10 1 (1) 0	0.2 0.5 0.9 0.6 0.9 0.6

## Data Availability

The data presented in this study are available on request from the corresponding author.

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
