# Peer review of "Evaluating the Protective Effect of Intravesical Bacillus Calmette-Guerin against SARS-CoV-2 in Non-Muscle Invasive Bladder Cancer Patients: A Multicenter Observational Trial"

_cancers, 2023, doi:10.3390/cancers15051618_

Round 1

Reviewer 1 Report

Very interesting clinical trials. But the title should be more representative of the work done. not sure whether it is suitable to be published in Cancers since the objective is protective role of BCG towards SARS-CoV2.

Author Response

We thank the reviewer for His/Her comment.

The title has been changed as follows: “ Evaluating the Protective Effect of Intravesical Bacillus Cal-mette-Guerin against SARS-CoV-2 in Non-Muscle Invasive Bladder Cancer Patients: A Multicenter Observational Trial”

We appreciate the consideration that the objective of the study is the protective role of BCG against covid. Nevertheless, it is important to emphasize that all patients enrolled in the study were patients with a diagnosis of bladder cancer and that the study aims to evaluate the role of intravesical administration of BCG, which is used as neoadjuvant therapy exclusively in patients diagnosed with urothelial carcinoma.

Reviewer 2 Report

The protective effect of intravesical Bacillus Calmette-Guerin (BCG) instillations against symptomatic SARS-CoV-2 infection in patients with high-risk and selected intermediate-risk non-muscle invasive bladder cancer has been evaluated. Consecutive patients treated with intravesical adjuvant therapy (BCG or chemotherapy) for NMIBC between January 2018 and December 2019 have been included in this study. Data, based on phone interviews, were collected between May and September 2020. There are numerous deficiencies in the study design. The patients were offered serological test, when molecular test was negative or they were never tested for SARS-CoV-2 infection. Serological test was not mandatory for those patients, thus, as addressed in the discussion, the actual number of seropositive patients is likely underestimated. When reporting SARS-CoV-2 infection-related symptoms (table 3) general flu-like symptoms in preceding 90 days were listed separately from the symptoms like fever, myalgia/arthralgia, headache, and symptoms requiring medical examination or hospitalization. Those symptoms, likely overlapping, were added as separate entities. Finally, the authors claim the novelty of their findings, which may have been novel at the time of the writing the first draft of this manuscript (as inferred from the statement in the introductory section " to date (May 2021))". Nevertheless, since than several publications reported results of the studies examining the effect of BCG instillations on symptomatic SARS-CoV-2 infection in patients with high-risk and selected intermediate-risk non-muscle invasive bladder cancer. Those references were omitted in this manuscript.

Author Response

The protective effect of intravesical Bacillus Calmette-Guerin (BCG) instillations against symptomatic SARS-CoV-2 infection in patients with high-risk and selected intermediate-risk non-muscle invasive bladder cancer has been evaluated. Consecutive patients treated with intravesical adjuvant therapy (BCG or chemotherapy) for NMIBC between January 2018 and December 2019 have been included in this study. Data, based on phone interviews, were collected between May and September 2020. There are numerous deficiencies in the study design. The patients were offered serological test, when molecular test was negative or they were never tested for SARS-CoV-2 infection. Serological test was not mandatory for those patients, thus, as addressed in the discussion, the actual number of seropositive patients is likely underestimated.

Reply:

We thank the reviewer for the time he/she spent reviewing our manuscript and for giving us the chance to improve it.

When reporting SARS-CoV-2 infection-related symptoms (table 3) general flu-like symptoms in preceding 90 days were listed separately from the symptoms like fever, myalgia/arthralgia, headache, and symptoms requiring medical examination or hospitalization. Those symptoms, likely overlapping, were added as separate entities.

Reply

We thank the reviewer for His/Her consideration, we apologize because perhaps table 3 lacks clarity.

The variables are not to be understood as exclusive, so the variable “Flu-like symptoms in the last 90 days” can include patients who developed symptoms such as fever, cough etc. This was specified in the table caption.

Finally, the authors claim the novelty of their findings, which may have been novel at the time of the writing the first draft of this manuscript (as inferred from the statement in the introductory section " to date (May 2021))". Nevertheless, since than several publications reported results of the studies examining the effect of BCG instillations on symptomatic SARS-CoV-2 infection in patients with high-risk and selected intermediate-risk non-muscle invasive bladder cancer. Those references were omitted in this manuscript.

We thank the reviewer for noting this lack. The discussion has been improved by including recent evidence regarding the role of BCG vaccination and instillations with BCG against covid.

Please see lines 185-253 and 266-273.

Reviewer 3 Report

This was an interesting report, describing the subject that demanded investigation, indeed. Authors made all possible efforts to collect as much information as possible, created very simple and relevant design of the study and made correct conclusions. Although the study did not present correlation between intravesical BCG and SARS-CoV-2 incidents it was attractive idea to perform this research. Of course, the retrospective nature of the study is limiting a factor. 

Author Response

We thank the reviewer for his/her kind comment. We agree that the retrospective nature of the study is a limitation. Nevertheless, it should be noted that the status of patients after enrolment (second wave) was evaluated prospectively.

Round 2

Reviewer 2 Report

The manuscript ID: cancers-2149164 by Hurle et al. has been considerably improved. The title of the manuscript in the revised version is far more appropriate. The relevant references have been included in the revised manuscript and the discussion modified accordingly.

Please, check and correct the references 3 and 5.

Author Response

We thank the reviewer for their comments. Ref 3 and 5 have been corrected
